# Renocardiac Effects of p-Cresyl Sulfate Administration in Acute Kidney Injury Induced by Unilateral Ischemia and Reperfusion Injury *In Vivo*

**DOI:** 10.3390/toxins15110649

**Published:** 2023-11-10

**Authors:** Carlos Alexandre Falconi, Fernanda Fogaça-Ruiz, Jéssica Verônica da Silva, Raquel Silva Neres-Santos, Carmen Lucía Sanz, Lia Sumie Nakao, Andréa Emília Marques Stinghen, Carolina Victoria Cruz Junho, Marcela Sorelli Carneiro-Ramos

**Affiliations:** 1Laboratory of Cardiovascular Immunology, Center of Natural and Human Sciences (CCNH), Federal University of ABC, Santo André 09210-170, SP, Brazil; a.falconi@ufabc.edu.br (C.A.F.); fernanda.f@aluno.ufabc.edu.br (F.F.-R.); j.veronica@aluno.ufabc.edu.br (J.V.d.S.); raquel.santos@ufabc.edu.br (R.S.N.-S.); 2Department of Basic Pathology, Universidade Federal do Paraná, Curitiba 81530-000, PR, Brazil; luciasanz@ufpr.br (C.L.S.); lia.nakao@ufpr.br (L.S.N.); 3Experimental Nephrology Laboratory, Basic Pathology Department, Universidade Federal do Paraná, Curitiba 81531-980, PR, Brazil; andreastinghen@ufpr.br; 4Institute for Molecular Cardiovascular Research (IMCAR), University Hospital RWTH Aachen, 52074 Aachen, Germany

**Keywords:** uremic toxins, renal diseases, cardiac diseases, cardiorenal axis

## Abstract

The precise mechanisms underlying the cardiovascular complications due to acute kidney injury (AKI) and the retention of uremic toxins like p-cresyl sulfate (PCS) remain incompletely understood. The objective of this study was to evaluate the renocardiac effects of PCS administration in animals subjected to AKI induced by ischemia and reperfusion (IR) injury. C57BL6 mice were subjected to distinct protocols: (i) administration with PCS (20, 40, or 60 mg/L/day) for 15 days and (ii) AKI due to unilateral IR injury associated with PCS administration for 15 days. The 20 mg/L dose of PCS led to a decrease in renal mass, an increase in the gene expression of Cystatin C and kidney injury molecule 1 (KIM-1), and a decrease in the α-actin in the heart. During AKI, PCS increased the renal injury biomarkers compared to control; however, it did not exacerbate these markers. Furthermore, PCS did not enhance the cardiac hypertrophy observed after 15 days of IR. An increase, but not potentialized, in the cardiac levels of interleukin (IL)-1β and IL-6 in the IR group treated with PCS, as well as in the injured kidney, was also noticed. In short, PCS administration did not intensify kidney injury, inflammation, and cardiac outcomes after AKI.

## 1. Introduction

Uremic toxins (UTs) are substances that accumulate in the body as a result of deteriorating renal function caused by acute or chronic kidney injury [1]. The gradual decline in kidney function leads to an accumulation of toxins normally cleared by the kidneys, resulting in uremia [2]. The presence of these toxins leads to cell and tissue damage and activates the immune system [3]. The classification of uremic compounds, as proposed by the European Uremic Toxins Work Group (EUTox), categorizes them into three main groups based on their molecular weight and their ability to be removed by dialysis membranes: water-soluble, medium-sized, and protein-bound uremic toxins (PBUTs) [4,5]. Among the protein-bound uremic toxins, there is a diverse pool of compounds with varying deleterious effects, already described in kidney dysfunction, endothelial dysfunction and thrombosis, and cardiac impairment [6]. p-cresyl sulfate (PCS), a product of tyrosine and phenylalanine metabolism by the microbiota, is approximately 95% bound to plasma albumin, and this complex is secreted by renal tubular proximal cells in normal conditions [7].

Acute kidney injury (AKI) is a serious and common condition that, while often potentially reversible, is associated with a significant increase in cardiovascular (CV) risk: AKI is associated with an 86% and a 38% increased risk of cardiovascular mortality and major cardiovascular events, respectively [8]. Also, mortality in AKI primarily results from non-renal organ failure that can occur because of distant effects [2]. The PBUTs serve as key mediators in the pathophysiological processes underlying AKI once their accumulation was already observed to induce (i) cardiac inflammation (via leucocyte activation both in vivo and *in vitro*) [9,10], (ii) cardiac oxidative stress and apoptosis [11], (iii) apoptosis in renal proximal tubular cells [12,13], and vascular dysfunction [14]. The increased systemic levels of PBUTs like PCS have already been correlated to cardiovascular (CV) risk in kidney disease patients [15]. This takes place in kidney patients and is not only attributed to the loss of kidney function but is also a result of an increase in the intestinal bacterial species, which are responsible for UT production. Also, the presence of intestinal dysbiosis in these patients significantly increases the likelihood of cardiovascular events [16].

Previous in vivo studies from our group have highlighted organ crosstalk between the kidney and heart during AKI. Using a unilateral renal ischemia and reperfusion (IR) injury model, we emphasized mainly the role of the inflammatory system in the progression of cardiac hypertrophy after 8 and 15 days of IR [17,18]. Additionally, our recent findings have demonstrated alterations in amino acid metabolism, increasing the availability of tyrosine and tryptophan (PCS and indoxyl sulfate metabolic precursors) after 8 and 15 days of IR [19]. Despite the critical implications of PBUTs alone, particularly PCS, in cardiac hypertrophy during AKI, no studies have gone far in studying the additional administration of PCS after an existing AKI. The hypothesis of this study is that the accumulation of the uremic compound PCS (through external administration), in association with an experimental AKI model, may lead to enhanced tissue damage in both kidney and heart when compared to AKI only. This study represents a pioneering effort to point out the potential effects of PCS during AKI, aiming to fill a critical gap in our understanding of the intricate renocardiac interplay. To test our hypothesis, we conducted two experimental sets to (i) determine the optimal dose of PCS capable of inducing renal alterations and (ii) combine PCS with the AKI model to evaluate the renal and cardiac effects.

## 2. Results

### 2.1. PCS Was Able to Induce Renal Damage and Cardiac Alterations with 20 mg/L Dose after 15 Days

Firstly, the macroscopic renal parameters of each experimental group were observed. The ratio of kidney weight/ tibia length (KW/TL) was significantly lower after the 20 mg/L dose, indicating renal atrophy when compared to vehicle-treated mice (Figure 1B). In addition, the mRNA levels of renal tissue damage markers were analyzed in both kidneys at the gene level. It was possible to observe a significant increase in the expression of Cystatin C and kidney injury molecule 1 (KIM-1) in the left kidney after 20 mg/L PCS administration for 15 days, together with the increase in KIM-1 expression in right kidney tissue after 20 and 40 mg/L PCS administration for 15 days (Figure 1C,D). Also, the 20 mg/L administration was revealed to provoke a body weight reduction after 15 days (Table 1). No differences were found regarding renal function after all concentrations of PCS administration for 15 days (Figure 1E,F).

To assess cardiac hypertrophy, the ratio of heart weight/ tibia length (HW/TL) was analyzed, together with the gene expression of α-actin, a marker of cardiac hypertrophy. The results suggest that even without alterations in heart weight (Figure 1G), a cardiac remodeling was observed once the PCS administration for 15 days in all studied doses was able to significantly downregulate the expression of α-actin in mice hearts (Figure 1H).

In short, a daily PCS administration of 20 mg/L for 15 days was able to induce kidney and heart alterations: a reduction in kidney weight (atrophy), an increase in expression of cystatin C in the left kidney, an increase in expression of KIM-1 in both kidneys, and a decrease in expression of α-actin (suggesting atrophy).

### 2.2. Unilateral Ischemia and Reperfusion (IR) Damaged the Left Kidney Tissue; However, PCS Administration Did Not Enhance the Injury after 15 Days

Following the IR injury and subsequent PCS administration for 15 days, we conducted plasma PCS level measurements (Figure 2B). While statistical differences among the experimental groups were not observed, some patterns emerged. Specifically, the sham and sham + PCS groups did not exhibit an increase in plasma PCS levels, which remained undetectable. However, a noticeable difference in PCS levels emerged after IR, underscoring the role of IR in PCS accumulation. Interestingly, this increase in PCS levels did not persist after PCS administration.

Moreover, no differences were found regarding renal function after all experimental groups’ administration for 15 days (Figure 2C,D). When assessing macroscopic renal parameters in each experimental group, no significant alterations in kidney ratios (LK/TL and RK/TL) were observed (Figure 2E,F and Table 2).

In terms of gene expression markers indicative of kidney damage, we analyzed Cystatin C, KIM-1, and neutrophil gelatinase-associated lipocalin (NGAL) in both kidneys. Notably, there were no discernible differences in Cystatin C expression in the injured kidney (Figure 2G). However, a significant effect of IR injury was evident when examining the contralateral kidney, as both the IR and IR + PCS groups exhibited a notable increase in Cystatin C expression compared to the sham and sham + PCS groups, respectively. Intriguingly, PCS administration did not exacerbate the deleterious effects of IR (Figure 2H). After evaluating KIM-1 expression in the left kidney, an increase was observed following 15 days of IR injury. However, when PCS administration was administered, either alone or in combination with IR, no significant alterations were found in the injured kidney. Similarly, in the right kidney across all analyzed groups, KIM-1 expression remained unchanged (Figure 2I,J). When analyzing NGAL, an increase in expression was noted in the left kidney after 15 days of IR injury. Interestingly, when combined with PCS administration, there was no further enhancement of NGAL expression (Figure 2K). Furthermore, there were no notable alterations in NGAL expression in the right kidney among all experimental groups (Figure 2L). It was possible to observe that IR itself damages the left kidney tissue once KIM-1 and NGAL mRNA levels are elevated after 15 days of IR. On the other hand, none of these markers is altered by the PCS administration and also not enhanced by the combination of IR + PCS.

### 2.3. Organic Anion Transporters (OATs) Are Modulated in the Injured Kidney by PCS Administration after Unilateral IR for 15 Days

Organic anion transporters (OATs) are essential to ensure homeostasis by proximal tubular secretion in the basolateral membrane of the renal proximal tubule cells [20,21]. Therefore, the affinity and expression of OATs are decisive for renal responses against substrates, metabolic residues, and uremic toxins [22].

The gene expression of OATs, specifically types 1 and 3, was analyzed in renal tissue. In relation to the left kidney (Figure 3A), the results indicated a decrease in the gene expression of OAT-1 in the IR + PCS groups compared to the sham + PCS group. The IR + PCS group maintained reduced gene levels when compared to the IR group, suggesting that PCS did not accentuate the negative modulation of this transporter in the kidney subjected to IR. Regarding OAT3, no changes were observed between the groups in the left kidney (Figure 3B). In the right kidney, the expression of OAT1 was decreased in the IR group compared to the sham group, but without further accentuation of the result with PCS administration (Figure 3C). On the other hand, in the right kidney, OAT3 was increased in the IR group compared to the sham group, and this effect was attenuated after PCS administration in the IR + PCS group (Figure 3D). In addition, serum IS levels were measured after 15 days of PCS administration and/or IR injury (Figure 3E). One can note that the IR injury alone was insufficient to elevate serum IS levels. However, the sham group treated with PCS showed a significantly higher level of IS when compared to the others. The IR + PCS administration also did not affect the serum IS levels.

### 2.4. PCS Combined with IR Prevented the Cardiac Hypertrophy Induced by Unilateral IR for 15 Days

When analyzing macroscopic parameters of the heart in the experimental groups, an increase in the ratio heart weight/tibia length (HW/TL) was observed after unilateral IR for 15 days, suggesting cardiac hypertrophy, already evidenced by our research group previously [17]. This alteration was not observed after the PCS administration and after PCS combined with the renal IR (Figure 4A). After analyzing the expression of α-actin, a marker for cardiac hypertrophy, the increase was confirmed after 15 days of unilateral IR. Curiously, after the combination of IR + PCS, it is possible to observe the prevention of an increase in α -actin expression (Figure 4B).

### 2.5. PCS Administration Did Not Enhance Inflammation Induced by Unilateral IR for 15 Days

When analyzing systemic secretion of interleukin (IL)-1β, it was possible to observe only a decrease after the combination of IR + PCS after 15 days, compared to the sham (Figure 5A), while IL-6 is upregulated after IR and after IR + PCS when compared to the sham group (Figure 5B). Analyzing the injured kidney tissue, it was possible to observe an increase in the expression of IL-1β and IL-6 after 15 days of IR injury when compared to the sham group. The IL-6 expression was also increased after 15 days of IR + PCS when compared to sham + PCS (Figure 5C,D). When looking at the inflammation in the heart tissue, it was possible to observe a decreased expression of IL-1β and an increased IL-6 expression in the IR group when compared to the sham group (Figure 5E,F). IL-1β was significantly increased when combined with IR + PCS (compared to IR), while IL-6 was decreased. In short, just PCS administration was not able to promote local and systemic inflammation, and the combination of IR + PCS was not able to induce more inflammation.

## 3. Discussion

In the present study, we provide for the first time insights into PCS’s effects on renal and cardiac health in the context of AKI. Surprisingly, while PCS is known to contribute to renocardiac deterioration in humans [23], our study yielded surprising results as PCS did not exacerbate the effects of renal IR injury, contrary to our expectations. The observed discrepancy between our mouse model and human studies regarding the impact of PCS on renal and cardiac health is noteworthy. To address this discrepancy, it is important to consider the inherent differences between our experimental mouse model and human studies. Mouse models, although valuable for investigating specific biological mechanisms, may not fully replicate the complexity of human physiology. This incongruity may be attributed to species-specific responses, genetic variations, and the controlled experimental conditions. Additionally, PCS exposure duration in our study, for example, may not perfectly mirror the clinical setting.

Our study shows that PCS at a dose of 20 mg/L for 15 days induced renal damage and cardiac alterations. PCS led to kidney atrophy and increased renal injury markers, and combining it with unilateral IR injury did not worsen kidney damage. In the left kidney, the organ submitted to ischemia and reperfusion, and the results for Cystatin C contradict the literature, which identifies PCS as a significant contributor to the development of renal inflammation and fibrosis [24]. In this regard, Sun et al. (2012) [24] highlighted the detrimental effects of PCS on the kidney, such as glomerulosclerosis, interstitial fibrosis, and the activation of the Renin Angiotensin Aldosterone System (RAAS), following the administration of this toxin, even in the absence of the IR process. On the other hand, the increase in KIM-1 and NGAL in the IR group was expected, as these sensitive markers are significantly correlated with renal deterioration in acute injury [25,26]. Despite their direct association with renal tissue damage, they also contribute to the reduction in glomerular filtration rate and inflammation [27]. As seen in Figure 2F,G,I, PCS administration did not increase the levels of these markers, indicating that PCS does not aggravate the kidney injury after IR. Furthermore, left kidney markers of inflammation (IL-6 and IL-1β) were elevated after 15 days of IR (Figure 5C,D), corroborating previous data from the lab [17,28]. Also, PCS does not aggravate the kidney’s local inflammation due to IR after 15 days. On the other hand, PCS was responsible for the increase in IL-6 in serum, suggesting systemic inflammation but not exacerbating the local inflammation in the kidney. Although IL-1β is highlighted in the literature as an extremely pro-inflammatory cytokine present in general cases of renal dysfunction [29,30], our findings did not identify increases in serum. Moreover, in contrast, L-1β appears to undergo negative modulation when IR and PCS administration are performed, suggesting a reduction in the overall inflammatory condition.

Next, the study examined gene expression of OAT 1 and 3 in renal tissue. The interaction between organic anion transporters and IS or PCS plays a crucial role in renal physiology and the elimination of uremic toxins. OATs are membrane proteins expressed in the kidneys that facilitate the transport of various molecules, including these sulfated compounds, from the blood into the urine [20]. OATs’ ability to influence the secretion and reabsorption of these sulfated compounds underscores their significance in maintaining a healthy renal environment. OAT1 expression did not alter the left kidney after IR, while unaffected by subsequent PCS administration, while OAT3 levels increased in the IR right kidney, but it was mitigated by PCS (Figure 3). OATs are vital for renal function, facilitating substrate secretion in proximal tubules. The results align with the literature, suggesting OAT1 reduction indicates acute renal insufficiency in AKI via protein kinase C activation [31,32]. However, our increased OAT3 expression contradicts prior findings in AKI rats [33]. AKI leads to tubular dysfunction, impacting compound reabsorption/excretion [34]. OATs 1 and 3 adapt to substrate demands, with phenolic compounds potentially influencing toxin elimination [35]. This was confirmed by IS measurement (Figure 3E), also accumulated in PCS-treated animals. Both OAT1 and OAT3 mediate the intake in the basolateral membrane of many drugs and compounds, raising the possibility of competition amongst them, leading to an alteration in the level of circulating UTs, which can explain the rise in the IS levels in the PCS-treated group [21,22] (Figure 3E), while variation of serum PCS cannot be seen. These findings have been critical in understanding the molecular mechanisms behind the renal handling of uremic toxins and their impact on kidney function.

Previous studies of the lab already demonstrated cardiac hypertrophy in the hearts of mice that passed through the IR surgery for 15 days [17,36,37], which was reaffirmed in the present study with the HW/TL and gene expression of α-actin. We were also able to observe what we dare to say is a preventive cardiac effect of PCS; once combined with IR, an increase in the HW/TL, together with a significant reduction in the gene expression of α-actin, was not observed. While some effects of PCS alone, such as increased renal inflammation and fibrosis, are known to contribute to cardiac hemodynamic alterations through the activation of RAAS or the overload caused by decreased glomerular filtration rate [38,39,40], our results imply a cardioprotective effect when combined with IR. Perhaps these findings reflect the apoptotic action of PCS in cardiomyocytes, counterbalancing the hypertrophic effect characteristic of AKI due to IR [9,11,41]. These results can be related to the local inflammation, presented in the present study by the increase in the IL-6 expression levels in the cardiac tissue after IR once inflammation strongly induces cardiac hypertrophy [42]. After 15 days of reperfusion and PCS administration, these expression levels also decreased; somehow, PCS attenuated the inflammation in the heart. These results also oppose other studies using the CKD model that identify PCS as a triggering agent in local and systemic inflammatory processes, capable of stimulating deleterious responses caused by both IR and the accumulation of uremic toxins in the circulation [24,39,41,43,44]. In the opposite direction, IR led to a decrease in cardiac IL-1β gene expression compared to the Sham group, while the IR + PCS group showed an increase in IL-1β compared to the IR group. The negative modulation of IL-1β in the IR group is a significant counterpoint that contradicts other findings, where the cytokine was described as cardioprotective [29,42,45]. These findings strongly suggest that the supplementation for 15 days of PCS is profitable in terms of hypertrophy and local inflammation when compared to the effects of the IR alone.

## 4. Conclusions

In summary, this study revealed that a 20 mg/L dose of PCS increased renal damage and cardiac tropism but did not potentiate gene expression changes in renal markers when combined with IR injury. When combined with IR, the PCS administration had no upregulation impact on cardiac tropism. OAT1 and OAT3 gene expressions displayed distinct patterns in the left kidney, while the right kidney exhibited altered OAT1 and OAT3 levels in response to IR. Additionally, IR reduced IL-1β but elevated IL-6, with PCS administration post-IR failing to further enhance the latter. These findings open avenues for future clinical research to better understand the intricate relationships between PCS and renal and cardiac dysfunction, offering potential insights into interventions to improve outcomes in clinical settings affected by uremic toxins or during the transition of AKI and CKD.

It is essential to acknowledge the limitations of our study. Firstly, our research was conducted in a controlled mouse model, and while this allowed for a detailed examination of PCS administration, it may not entirely mirror the complexity of human pathophysiology. This divergence underscores the challenge of directly extrapolating our findings to clinical scenarios. Secondly, the dosage and duration of PCS exposure in our study do not precisely mimic the clinical context in humans. Clinical exposures to PCS can vary considerably, depending on factors such as diet, kidney function, gut health, and metabolism. Additionally, it is essential to recognize that the interplay of multiple variables in clinical settings, including patient diversity, comorbidities, and multifactorial influences, can yield different outcomes not accounted for in our controlled experiment. Furthermore, to gain a better understanding of the underlying mechanisms, it would be valuable to employ specific blockers of organic anion transporters (OATs), such as probenecid. Previous studies have demonstrated the utility of probenecid in investigating endothelial dysfunction and apoptosis [31,46,47]. Its application in our research can shed light on the intricate pathways through which PCS influences renal and cardiac health.

## 5. Materials and Methods

### 5.1. Animal Procedures

The experiments were carried out under Brazilian Federal Law No. 11794 of 8 October 2008, which regulates the use of animals in scientific experimentation, and in conformity with the institutional ethical guideline of the Federal University of ABC (under number 6548291020). Male C57BL/6 mice were used, aged between 6 and 8 weeks, weighing between 20 and 30 g. All animals were placed in cages containing one animal per cage, with an artificial light/dark cycle of 12 h, at a constant ambient temperature of 25 °C, and with supplemental water and food always available.

### 5.2. Acute Kidney Injury Protocol of Unilateral Ischemia-Reperfusion (IR)

The protocol of AKI induction by unilateral renal IR was performed as described before [17,18,28]. In brief, the animals were anesthetized with xylazine and ketamine (10 mg/kg and 80 mg/kg, respectively, dissolved in a 0.9% saline solution) via intraperitoneal injection (i.p.). The abdominal cavity was opened to access the renal pedicle, and the left renal artery was clamped for 60 min. After ischemia, the clamp was removed, and the abdominal organs were repositioned. Peritoneal and skin sutures were performed using nylon 6-0 and silk 4-0 threads, respectively.

### 5.3. p-Cresyl Sulfate (PCS) Administration

p-cresyl was kindly provided by Dr. Andrea Stinghen from the Federal University of Curitiba. The animals received PCS under two conditions. In the first one, mice were divided into four groups: a vehicle group (which was treated with 0.9% saline solution i.p.) and three experimental groups, which received concentrations of 20 mg/L, 40 mg /L, and 60 mg/L of PCS, i.p., added with 0.9% saline solution (Figure 1A). The doses were based on the Eutox Database, and the time of administration was chosen according to a previous study from our group. In the second condition, the PCS administration was combined with renal unilateral IR. They were divided into four other groups: sham (animals that underwent the surgical procedure, except for the occlusion of the renal pedicle and reperfusion, and received daily vehicle administration for 15 days, i.p.); sham + PCS (same as before but received daily administration of PCS at a dose of 20 mg/L from the day after the surgery, for 15 days, i.p.); IR (underwent the surgical procedure of occlusion of the left renal pedicle for 60 min; during the 15-day reperfusion they received daily vehicle administration, i.p.) and IR + PCS (underwent the surgical procedure of occlusion of the left renal pedicle for 60 min; during the 15-day reperfusion they received daily administration of PCS at a dose of 20 mg/L from the day after the surgery, i.p.). After the 15th day of reperfusion, the animals were euthanized. Blood, heart, kidneys, and tibia were collected and kept in a freezer at −80 °C until they were processed for further analyses (Figure 2A).

### 5.4. Serum Measurements

After removal, the blood from the mice was placed in a microtube and centrifuged for 10 min at 10,000× *g* and 4 °C. The serum was removed with a micropipette and transferred to another clean microtube until the time of dosage. They were kept in a freezer at −80 °C. Serum urea and creatinine dosages (mg/dL) were performed by colorimetric analysis using Urea CE kits and Creatinine kits (LabTest Diagnóstica SA), respectively, according to the manufacturer’s instructions. Also, to measure cytokines in serum, the Milliplex Mouse Cytokine/Chemokine Magnetic Bead Panel (Merck Millipore, Amsterdam, Holland) was used according to the manufacturer’s instructions. The cytokines analyzed were IL-6 and IL-1β. Serum cytokine data were obtained in pg/mL by calculating the median fluorescence intensity using a 5-parameter logarithmic curve.

### 5.5. Assessment of Cardiac and Renal Tropism

The tropism of the heart and kidneys was evaluated by morphometric analysis given by the ratio between the weight of the organ and the length of the tibia. Expression analysis of cardiac hypertrophy markers, α-actin, was also carried out.

### 5.6. Gene Expression Analysis by Real-Time Polymerase Chain Reaction (RT-PCR)

The RNA was obtained from tissue samples using Trizol^®^ (50–100 mg of the sample), as per the manufacturer’s instructions. Then, all samples were stored at −80 °C until the moment of quantification in a spectrophotometer at the following wavelengths: 260 nm and 280 nm in the NanoDrop Lite Spectrophotometer—Thermo Scientific equipment (Waltham, MA, USA). The extracted RNA was submitted to a reverse transcription reaction using the High-Capacity cDNA Reverse Transcription—Applied Biosystems kit according to the manufacturer’s recommendations. Results were expressed based on the ratio of the mRNA of each gene of interest to cyclophilin A mRNA levels, and samples from independent experiments were processed in duplicate. The primers were designed using the Primer-BLAST software (https://blast.ncbi.nlm.nih.gov/Blast.cgi, National Center for Biotechnology Information—NCBI) and are available in Table 3. The data were expressed by fold change considering the housekeeping cyclophilin A.

### 5.7. Plasmatic Measurement

PCS and IS were measured using HPLC. In brief, plasma samples were processed as described [48]. Briefly, 50 μL of plasma and 2.5 μL 4-ethylphenol (12.4 mM) as an internal standard were diluted with water to 180 μL and heated (95 °C, 30 min). After 10 min in ice, samples were centrifuged (13,000 rpm in bench centrifuge, 4 °C, 20 min), and the supernatant was ultrafiltered (11,000 rpm in bench centrifuge, 4 °C, 30 min) with a 30 kDa cutoff membrane (Amicon Ultra, Millipore). The ultrafiltrate (10 μL) was injected. A four-point standard curve was constructed by mixing varying amounts of p-cresyl sulfate (0–203 pmol) and indoxyl sulfate (0–122 pmol) with control plasma, internal standard, and water to reach 180 μL.

### 5.8. Statistical Analysis

Data are stated as the mean ± standard deviation (SD). Statistical significance was accomplished by one-way ANOVA, with Dunnet’s or Tukey’s multiple comparison test, when appropriate. The statistical analysis was performed with the GraphPad Prism v9.0 (GraphPad Software Inc., San Diego, CA, USA) software program. Differences with *p*-values lower than 0.05 were considered statistically significant.

## Figures and Tables

**Figure 1 toxins-15-00649-f001:**
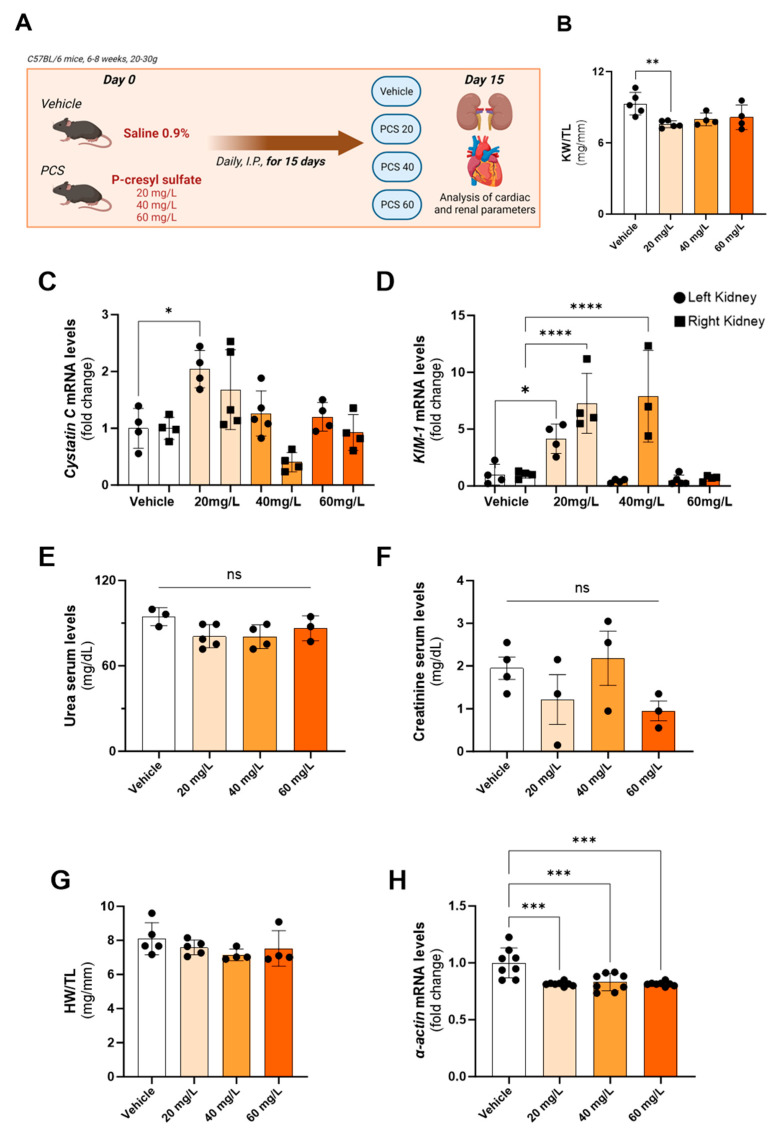
Effect of p-cresyl sulfate (PCS) in the cardio-renal axis (cardiac and renal tissue) of mice after PCS administration for 15 days. (**A**) Experimental design of PCS administration using 20, 40, or 60 mg/L, i.p., daily for 15 days. (**B**) Kidney weight/tibia ratio (mg/mm). (**C**) Renal Cystatin C and (**D**) kidney injury molecule 1 (KIM-1) mRNA levels of the experimental groups. Round dots represent left kidney tissue, while square represents right kidney tissue. (**E**) Urea and (**F**) creatinine serum concentration (mg/dL) after PCS administration after 15 days in different concentrations. (**G**) Heart weight/tibia ratio (mg/mm) of the experimental groups. (**H**) Cardiac α-actin mRNA levels of the experimental groups. Data are expressed as the mean ± standard deviation (SD). * *p* < 0.05, ** *p* < 0.01, *** *p* < 0.001 and **** *p* < 0.0001 for the respective shown groups after one-way ANOVA followed by Dunnett’s multiple comparison test.

**Figure 2 toxins-15-00649-f002:**
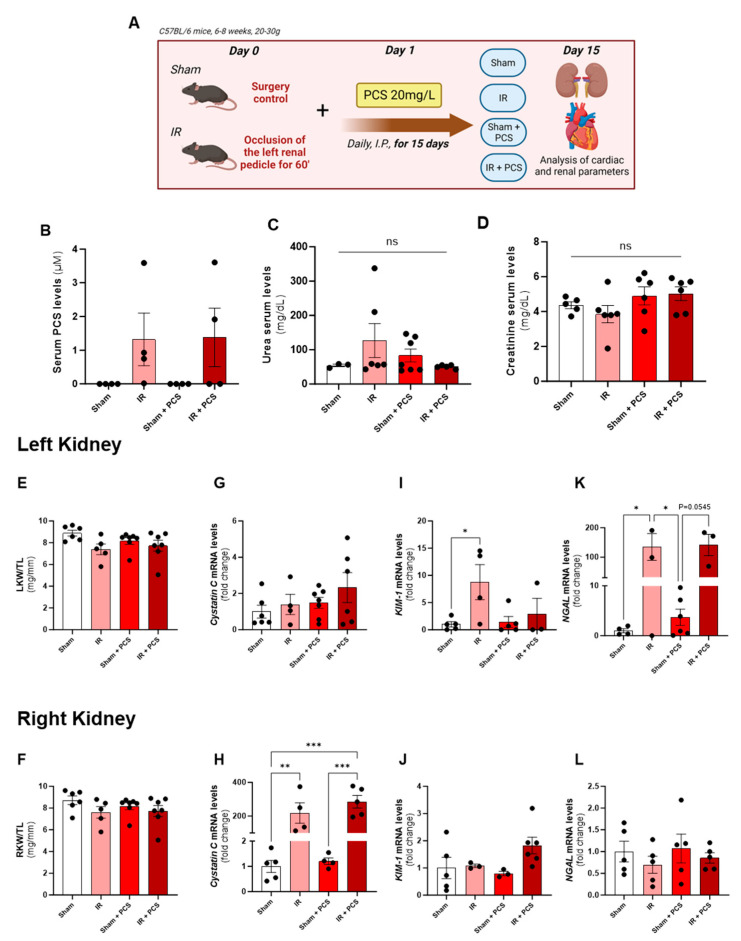
Renal effects of p-cresyl sulfate (PCS) after acute kidney injury (AKI) of mice induced by unilateral renal ischemia and reperfusion (IR) and PCS administration after 15 days. (**A**) Experimental design of combination of left IR for 60 min and PCS 20 mg/L administration. (**B**) Serum levels of PCS in the experimental groups. (**C**) Urea and (**D**) creatinine serum concentration (mg/dL) after renal IR and PCS administration after 15 days. (**E**) Left (injured) and (**F**) right (counterbalance) kidney weight/tibia ratio (mg/mm) of the experimental groups. (**G**) Left and (**H**) right kidney Cystatin C mRNA levels of the experimental groups. (**I**) Left and (**J**) right kidney injury molecule 1 (KIM-1) mRNA levels of the experimental groups. (**K**) Left and (**L**) right neutrophil gelatinase-associated lipocalin (NGAL) mRNA levels of the experimental groups. Data are expressed as the mean ± standard deviation (SD). * *p* < 0.05, ** *p* < 0.01, and *** *p* < 0.001 for the respective shown groups after one-way ANOVA followed by Tukey’s multiple comparison test.

**Figure 3 toxins-15-00649-f003:**
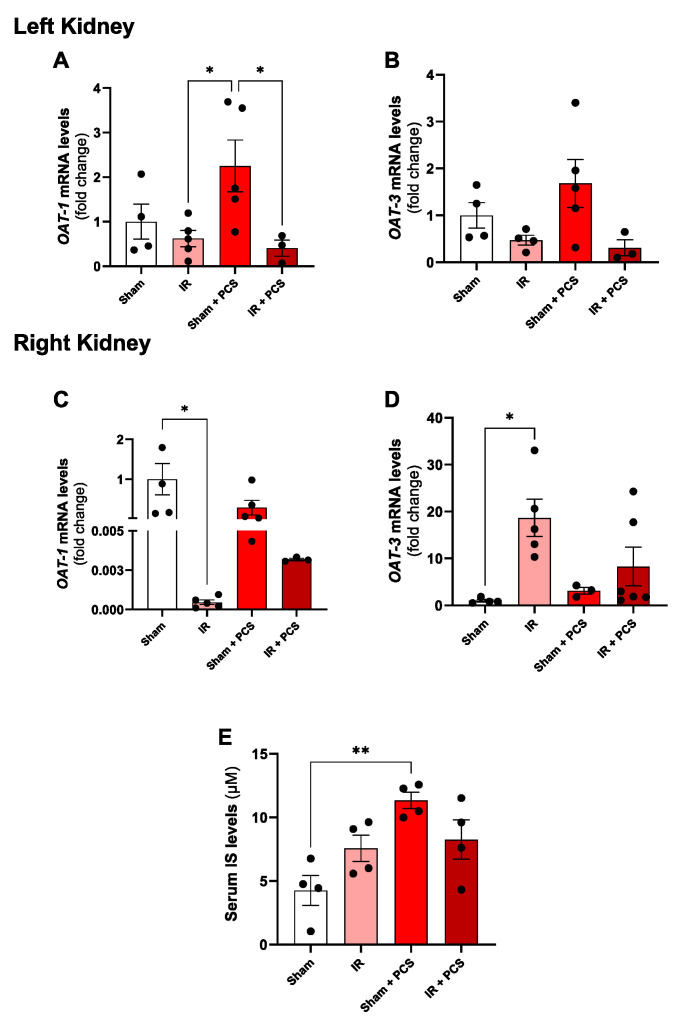
Effects of p-cresyl sulfate (PCS) in gene expression of organic anion transporters (OAT) and indoxyl sulfate (IS) levels after acute kidney injury (AKI) of mice induced by unilateral renal ischemia and reperfusion (IR) injury and PCS administration after 15 days. Left kidney (**A**) OAT1 and (**B**) OAT3mRNA levels of the experimental groups. Right kidney (**C**) OAT1 and (**D**) OAT3 mRNA levels of the experimental groups. (**E**) Serum levels of IS in the experimental groups. Data are expressed as the mean ± standard deviation (SD). * *p* < 0.05 and ** *p* < 0.01 for the respective shown groups after one-way ANOVA followed by Tukey’s multiple comparison test.

**Figure 4 toxins-15-00649-f004:**
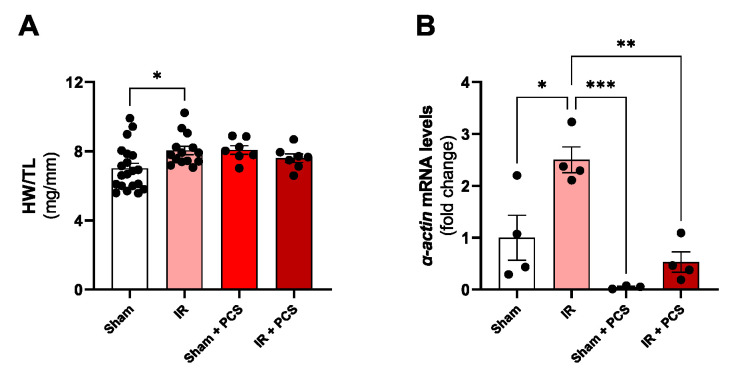
Cardiac effects of p-cresyl sulfate (PCS) after acute kidney injury (AKI) of mice induced by unilateral renal ischemia and reperfusion (IR) and PCS administration after 15 days. (**A**) Heart weight/tibia ratio (mg/mm) and (**B**) cardiac α-actin mRNA levels of the experimental groups. Data are expressed as the mean ± standard deviation (SD). * *p* < 0.05, ** *p* < 0.01, and *** *p* < 0.001 for the respective shown groups after one-way ANOVA followed by Tukey’s multiple comparison test.

**Figure 5 toxins-15-00649-f005:**
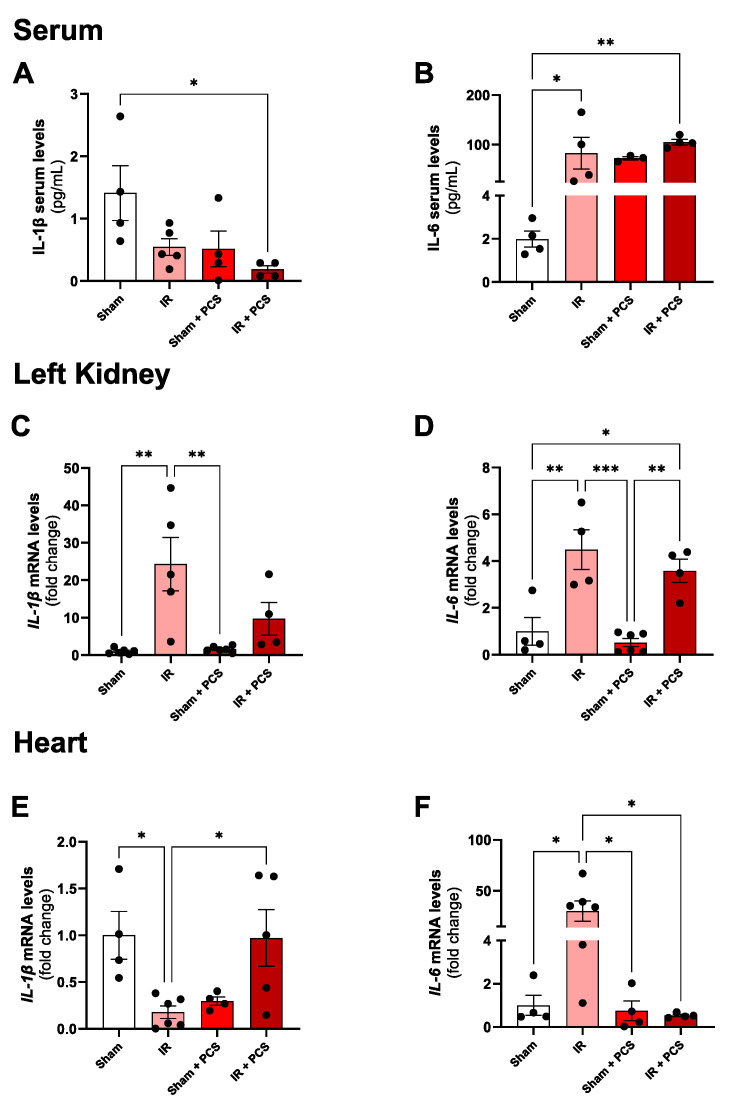
Inflammatory panel measured in serum, left kidney, and heart after acute kidney injury (AKI) of mice induced by unilateral renal ischemia and reperfusion (IR) and p-cresyl sulfate (PCS) administration after 15 days. Systemic interleukin (IL) 1β (**A**) and IL-6 (**B**). Left kidney mRNA expression of IL-1β (**C**) and IL-6 (**D**) of the experimental groups. Cardiac mRNA expression of IL-1β (**E**) and IL-6 (**F**) of the experimental groups. Data are expressed as the mean ± standard deviation (SD). * *p* < 0.05, ** *p* < 0.01, and *** *p* < 0.001 for the respective shown groups after one-way ANOVA followed by Tukey’s multiple comparison test.

**Table 1 toxins-15-00649-t001:** Morphometric data on body, heart, and kidney weight and tibia size of mice after PCS administration for 15 days. Experimental n is expressed in brackets. Data are expressed as mean ± SD. * *p* < 0.05, ** *p* < 0.01, and *** *p* < 0.001 vs. vehicle after one-way ANOVA followed by Dunnett’s multiple comparison test.

Parameter	Vehicle (5)	PCS 20 mg/L (5)	PCS 40 mg/L (4)	PCS 60 mg/L (4)
**BW** (g)	25.85 ± 2.13	20.35 ± 1.63 ***	25.47 ± 0.93	26.33 ± 1.36
**HW** (g)	0.1332 ± 0.015	0.1172 ± 0.008	0.1105 ± 0.007	0.1173 ± 0.015
**LK** (g)	0.1514 ± 0.019	0.1240 ± 0.011 *	0.1155 ± 0.008 *	0.0111 ± 0.022
**RK** (g)	0.1556 ± 0.014	0.1288 ± 0.008 **	0.1315 ± 0.01 *	0.1303 ± 0.014 *
**TL** (mm)	15.46 ± 0.20	15.45 ± 0.33	15.44 ± 0.24	15.55 ± 0.32

**Table 2 toxins-15-00649-t002:** Morphometric data on body, heart, and kidney weight and tibia size of mice after unilateral renal IR and PCS administration after 15 days. Experimental n is expressed in brackets. Data are expressed as mean ± SD. * *p* < 0.05 vs. Sham after one-way ANOVA followed by Tukey’s multiple comparison test.

Parameter	Non-Treated		+PCS	
	Sham (6)	IR (6)	Sham (7)	IR (7)
**BW** (g)	24.49 ± 3	24.25 ± 1.34	24.14 ± 1.27	23.47 ± 1.23
**HW** (g)	0.1414 ± 0.021	0.466 ± 0.173 *	0.1350 ± 0.010	0.1246 ± 0.009
**LK** (g)	0.1612 ± 0.016	0.1267 ± 0.02 *	0.1364 ± 0.014	0.1267 ± 0.002
**RK** (g)	0.1612 ± 0.028	0.1619 ± 0.027	0.1498 ± 0.017	0.1591 ± 0.003
**TL** (mm)	16.72 ± 0.31	19.97 ± 0.53	16.71 ± 0.19	16.40 ± 0.51

**Table 3 toxins-15-00649-t003:** Primers used for real-time PCR.

Gene	Sequence Forward	Sequence Reverse
*Cyclophilin A*	AGCATACAGGTCCTGGCATC	AGCTGTCCACAGTCGGAAAT
*Cystatin C*	GCGTACCACAGCCGCGCCAT	TGGGGCTGGTCATGGAAAGGA
*KIM-1*	TGGCACTGTGACATCCTCAGA	GCAACGGACATGCCAACATA
*NGAL*	CACCACGGACTACAACCAGTTCGC	TCAGTTGTCAATGCATTGGTCGGTG
*OAT1*	CCATCGTGACTGAGTGGAAC	TGTCCGCCAGGTAGCCAAAC
*OAT3*	CTTCAGAAATGCAGCTCTTG	ACCTGTTTGCCTGAGGACTG
*α-actin*	GGCAAGATGAGAGTGCACAA	CGGAGAATGATGGTCCAGAT
*IL-6*	CTCTGCAAGAGACTTCCATCC	CTCTGTGAAGTCTCCTCTCCG
*IL1-β*	AGTTGACGGACCCCAAAAGA	GCTCTTGTTGATGTGCTGCT

## Data Availability

Data are available on request.

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
