# Peer review of "Renocardiac Effects of p-Cresyl Sulfate Administration in Acute Kidney Injury Induced by Unilateral Ischemia and Reperfusion Injury In Vivo"

_toxins, 2023, doi:10.3390/toxins15110649_

Round 1

Reviewer 1 Report

Comments and Suggestions for Authors

This paper assessed the effects of PCS treatment on renal cardiac function in animals with acute kidney injury induced by ischemia-reperfusion (IR) injury. The paper concluded that PCS treatment does not exacerbate renal injury, inflammation, or cardiac prognosis after AKI. Overall, this is a well-written paper. The authors strongly this paper to be accepted after minor modification. Some of the suggestions are as below:

(1) The supplementary material seems to be only one page; could it be placed in the body of the text?

(2) Please avoid lumped references, such as [13-15].

(3) Could the authors enrich the Introduction section?

(4) The authors are suggested to mention the structure of this paper at the end of Introduction section.

Comments on the Quality of English Language

The writing is acceptable.

Author Response

Reviewer 1:

This paper assessed the effects of PCS treatment on renal cardiac function in animals with acute kidney injury induced by ischemia-reperfusion (IR) injury. The paper concluded that PCS treatment does not exacerbate renal injury, inflammation, or cardiac prognosis after AKI. Overall, this is a well-written paper. The authors strongly this paper to be accepted after minor modification. Some of the suggestions are as below:

The supplementary material seems to be only one page; could it be placed in the body of the text?

Answer: Thanks for the suggestion. We added the supplementary figure in the body of the text. They were included in figure 1 and 2 and are now table 1 and 2.

Please avoid lumped references, such as [13-15].

Answer: Thanks for the suggestion. We have now avoided lumped references.

Could the authors enrich the Introduction section?

Answer: Thank you for the suggestion. As reviewer 2 has suggested the same, enriched the introduction section. We added some information about acute kidney injury (AKI) and correlations to cardiovascular risk, previously missing, including new references.

The authors are suggested to mention the structure of this paper at the end of Introduction section.

Answer: Thank you for the suggestion. We've included a statement at the end of introduction (line 66-69).

Reviewer 2 Report

Comments and Suggestions for Authors

The hypothesis of this study is that the accumulation of the uremic compound p-cresyl sulfate (PCS), in association with an experimental model of acute renal failure (AKI), can lead to tissue damage in both kidney and heart. In particular, the renocardiac effects of PCS treatment were evaluated both in animals undergoing ischaemia and reperfusion (IR)-induced AKI and in animals without renal damage.  The authors showed that PCS treatment in the absence of AKI induced renal damage and cardiac changes, whereas in the presence of AKI, induced by IR, it did not worsen renal and cardiac damage.

The manuscript is well presented but the following aspects should be revised by the authors:

- It should be stated in the title that the study is based on in vivo experiments to make the article more interesting for the readers;

- In line 35, it is not necessary to repeat the same quote twice, it would be sufficient to insert it once at the end of the sentence;

- It should be stated in the introduction that uremic toxins, including PCS, accumulate in patients with chronic and acute renal failure not only due to their failure to be excreted by the kidney but also due to an increase in the intestinal bacterial species responsible for their production. (refer to article: doi: 10.3390/genes14061257.)

- Another key aspect to consider when discussing uremic toxins associated with kidney damage is intestinal dysbiosis. Indeed, it is well known that in subjects with kidney damage, the development of intestinal dysbiosis occurs already in the early stages of the disease. The condition of gut dysbiosis in these patients promotes the onset of cardiovascular events. The authors are requested to elaborate on this topic in the introduction or discussion. (Refer to article: doi: 10.1007/s00394-018-1785-z.)

Author Response

Review 2:

The hypothesis of this study is that the accumulation of the uremic compound p-cresyl sulfate (PCS), in association with an experimental model of acute renal failure (AKI), can lead to tissue damage in both kidney and heart. In particular, the renocardiac effects of PCS treatment were evaluated both in animals undergoing ischemia and reperfusion (IR)-induced AKI and in animals without renal damage.  The authors showed that PCS treatment in the absence of AKI induced renal damage and cardiac changes, whereas in the presence of AKI, induced by IR, it did not worsen renal and cardiac damage.

The manuscript is well presented but the following aspects should be revised by the authors:

It should be stated in the title that the study is based on in vivo experiments to make the article more interesting for the readers.

Answer: Thank you for the suggestion. Indeed, the inclusion of the idea of animal experimentation already in the title gives a better perception of the study itself. We modified the tittle based in your suggestion together with Reviewer 3’s one. The new title is: “Renocardiac effects of p-cresyl sulfate administration in acute kidney injury induced by unilateral ischemia and reperfusion injury in vivo”.

In line 35, it is not necessary to repeat the same quote twice, it would be sufficient to insert it once at the end of the sentence.

Answer: Thank you for the comment, we reformulated the sentence in the introduction section:

(…) and this complex is secreted by renal tubular proximal cells in normal condition. When decrease the renal function decreases, as the acute kidney injury (AKI) or the chronic kidney disease (CKD) (…)

It should be stated in the introduction that uremic toxins, including PCS, accumulate in patients with chronic and acute renal failure not only due to their failure to be excreted by the kidney but also due to an increase in the intestinal bacterial species responsible for their production. (refer to article: doi: 10.3390/genes14061257.)

Another key aspect to consider when discussing uremic toxins associated with kidney damage is intestinal dysbiosis. Indeed, it is well known that in subjects with kidney damage, the development of intestinal dysbiosis occurs already in the early stages of the disease. The condition of gut dysbiosis in these patients promotes the onset of cardiovascular events. The authors are requested to elaborate on this topic in the introduction or discussion. (Refer to article: doi: 10.1007/s00394-018-1785-z.).

Answer: Thank you for the suggestion, we reformulated the introduction section in order to include the information from both comments:

“The increased systemic levels of PBUTs, as PCS, has already been correlated to cardiovascular (CV) risk in kidney disease patients (15).  This take place in kidney patients is not only attributed to the loss of kidney function, but also as a result of an increase in the intestinal bacterial species, which are responsible for UTs production. Also, the presence of intestinal dysbiosis in these patients significantly increases the likelihood of cardiovascular events (16)”.

Reviewer 3 Report

Comments and Suggestions for Authors

Comments on the Quality of English Language

Dear Editor,

I recommend to greatly revise the use of the English language.

All the best for you and the toxins

Author Response

Reviewer 3:

This study presents a comprehensive investigation into the impact of pcresyl sulfate on the renocardiac system in mice with acute renal injury. The use of experimental animal models in research could significantly enhance our understanding of analogous conditions in humans. For comments, please refer to the following section.

Major comments

The term ‘treatment’ creates some confusion already in the title. Searching a proper definition, you might find: In the context of medicine, "treatment" refers to the therapeutic actions or interventions undertaken by healthcare professionals to alleviate, manage, or cure a medical condition or disease. It is thus suggested to replace ‘treatment’. Administration or similar might do the job.

Answer: Thank you for the comment. Indeed, the use of the term treatment gives the wrong idea about the compound. We substituted the term “treatment” for “administration” in the whole manuscript.

Discussion 1. Para: If you state that the results from mice are not in agreement with those from humans, then please discuss the use of the experimental model.

Answer: Thank you for the suggestion. Indeed, this discussion was missing in our manuscript. We included a discussion concerning the point raised in the results section:

“Surprisingly, while PCS is known to contribute to renocardiac deterioration in humans [23], our study yielded surprising results as PCS did not exacerbate the effects of renal IR injury, contrary to our expectations. The observed discrepancy between our mouse model and human studies regarding the impact of PCS on renal and cardiac health is noteworthy. To address this discrepancy, it is important to consider the inherent differences be-tween our experimental mouse model and human studies. Mouse models, although valuable for investigating specific biological mechanisms, may not fully replicate the complexity of human physiology. This incongruity may be attributed to specific responses, genetic variations, and the controlled experimental conditions. Additionally, PCS exposure duration in our study, for example, may not perfectly mirror the clinical setting”.

Use of the English language needs being improved. Two examples: (a) cardiac trophism; tropism is meant? (b) Lately, the analysis … Do you mean finally? One example for phrasing: l. 146: It is possible to observe that the IR injury by itself wasn’t enough to increase the serum levels of IS. How about: One can note that the IR injury alone was insufficient to elevate serum IS levels.

Answer: Thank you for the comment. The sentence suggested was included in the manuscript. English language was also corrected and improved.

Limitations of the study: Maybe here or in the Discussion section discuss to which extent the results can sensefully be translated to humans.

Answer: Thank you for the suggestion. A paragraph including the suggestion as included in the end of the conclusion section:

“It is essential to acknowledge the limitations of our study. Firstly, our research was conducted in a controlled mouse model, and while this allowed for a detailed examina-tion of PCS administration, it may not entirely mirror the complexity of human patho-physiology. This divergence underscores the challenge of directly extrapolating our find-ings to clinical scenarios. Secondly, the dosage and duration of PCS exposure in our study not precisely mimic the clinical context in humans. Clinical exposures to PCS can vary considerably, depending on factors such as diet, kidney function, gut healthy and metabolism. Additionally, it is essential to recognize that the interplay of multiple variables in clinical settings, including patient diversity, comorbidities, and multifactorial influences, can yield different outcomes not accounted for in our controlled experiment. Furthermore, to gain a better understanding of the underlying mechanisms, it would be valuable to employ specific blockers of organic anion transporters (OATs), such as probenecid. Previous studies have demonstrated the utility of probenecid in investigating endothelial dysfunction and apoptosis [31, 43, 44]. Its application in our research could shed light on the in-tricate pathways through which PCS influences renal and cardiac health”.

Minor comments

  1. 20: ….. to AKI didn’t not exacerbates ….
  2. 62: This data evidence the renal … Correct?
  3. 75: To access cardiac hypertrophy in this first moment, it was

analyzed the ratio heart weight/ tibia length (HW/TL),

  1. 124: In short, it was possible to observe that IR itself damage the left

kidney tissue,

  1. 134: for the ‘beauty’ please, ascending numbers, throughout
  2. 160: parameters?
  3. 287: Lately, the analysis of another heart parameters would clarify the

cardiac role of PCS in this model. Sentence correct?

  1. 337: The trophism of the heart and kidneys were evaluated by… was
  2. 354 Plasmatic uremic toxins measurement. What do you want to

say? Plasmatic measurements?

Answer: Thank you for the minor comments. All the points raised by reviewer was corrected and presented in new version of the manuscript. 

Round 2

Reviewer 3 Report

Comments and Suggestions for Authors

Thanks for considering the comments.

Please, replace the term therapy also in the title by administration.

Comments on the Quality of English Language

While the English is comprehensible, the phrasing often hurts.

Author Response

Dear Reviewer, 

Thank you for the suggestion. The term in the title was replaced for "administration".